# Candidate Genes and Pathways in Cervical Cancer: A Systematic Review and Integrated Bioinformatic Analysis

**DOI:** 10.3390/cancers15030853

**Published:** 2023-01-30

**Authors:** Marjanu Hikmah Elias, Srijit Das, Nazefah Abdul Hamid

**Affiliations:** 1Department of Basic Medical Sciences I, Faculty of Medicine & Health Sciences, Universiti Sains Islam Malaysia, Nilai 71800, Malaysia; 2Department of Human & Clinical Anatomy, College of Medicine and Health Sciences, Sultan Qaboos University, Muscat 123, Oman

**Keywords:** cervical cancer, differentially expressed gene, molecular pathway, gene ontology

## Abstract

**Simple Summary:**

Cervical cancer is the fourth most common cancer among women worldwide. Although many recommendations on the screening, diagnosis, and treatment of cervical cancer have been established, no comprehensive molecular mechanism for cervical cancer has been determined. Hence, this systematic review and integrated bioinformatic analysis provides new insight into the key genes and pathways involved in the pathogenesis of cervical cancer and, thus, can be beneficial for developing better screening and treatment strategies for cervical cancer.

**Abstract:**

Cervical cancer is the leading cause of cancer-related death among women in developing countries. However, no comprehensive molecular mechanism for cervical cancer has been established, as many studies were small-cohort studies conducted with small sample sizes. A thorough literature search was performed using the PubMed, Scopus, EBSCOhost, and Science Direct databases. Medical Subject Heading (MeSH) terms such as “Uterine Cervical Neoplasms” and “gene expression” were used as the keywords in all fields. A total of 4027 studies were retrieved, and only clinical studies, which used the microarray method to identify differentially expressed genes (DEGs) in the cervical tissue of cervical cancer patients, were selected. Following the screening, 6 studies were selected and 1128 DEGs were extracted from the data. Sixty-two differentially expressed genes from at least two studies were selected for further analysis by DAVID, STRING, and Cytoscape software. In cervical cancer pathogenesis, three significant clusters with high intermolecular interactions from the Protein–Protein Interaction (PPI) network complex revealed three major molecular mechanisms, including cell signaling, cell cycle, and cell differentiation. Subsequently, eight genes were chosen as the candidate genes based on their involvement in the relevant gene ontology (GO) and their interaction with other genes in the PPI network through undirected first neighbor nodes. The present systematic review improves our understanding of the molecular mechanism of cervical cancer and the proposed genes that can be used to expand the biomarker panel in the screening for cervical cancer. The targeted genes may be beneficial for the development of better treatment strategies.

## 1. Introduction

Cervical cancer is the fourth most common cancer in women worldwide, with an estimated incidence of 570,000 and 311,000 related deaths in 2018 [1]. Although the incidence of cervical cancer has declined in the United States, it remains the leading cause of cancer-related death in developing countries [2]. Thus, many recommendations on the screening, diagnosis, and treatment of cervical cancer have been established [2,3,4]. A recent recommendation on multiple-agent chemotherapy regimens that include targeted therapies and immunotherapy regimens in combination with the existing first- and second-line treatment options has improved the outcomes in some cervical cancer patients [3]. However, cervical cancer is a multifactorial disease that involves complex pathophysiology and molecular pathogenesis. Thus, further studies are still crucial to providing comprehensive information on the mechanism of cervical cancer pathogenesis that is critical in developing better screening, diagnosis, and treatment of cervical cancer.

Several molecular mechanisms have been proposed for cervical cancer involving genomic variations [5], regulation of mRNA expression [6], and epigenetic changes [7]. Various pathways have been identified to be involved in cervical cancer through the regulation of gene expression including *MAPK, mTOR, PI3K-Akt*, the *Ras* signaling pathway, immune response, inflammation, DNA synthesis, cell proliferation, and many others [8,9]. Identifying the DEGs and their key pathways and functions can improve our understanding of the molecular mechanisms that occur in cervical cancer pathogenesis. Therefore, many studies on gene-expression profiling have been conducted on cervical cancer patients, and hundreds of DEGs have been identified. From these studies, many potential biomarkers have been proposed, such as *FBL1*, *ANT3* [10], *RBBP6* [11], *TMEM45A*, *SERPINB5, p16INK4A* [12], and many more. However, these studies only focused on a single cohort from different populations, with a small sample size, and employed different methods to identify the DEGs. Thus, these individual results could not robustly represent the pathogenesis of cervical cancer due to various biases. Hence, this systematic review was performed to identify the pattern of gene expression, functions, interactions, and critical pathways with minimal bias in the cervical tissue of cervical cancer patients across populations via integrated bioinformatic analysis.

## 2. Materials and Methods

### 2.1. Search Strategy

The systematic review was conducted according to the PRISMA guidelines. This systematic review was registered with OpenScience Framework. A comprehensive literature search was carried out using the PubMed, Scopus, EBSCOhost, and Science Direct databases. Related research papers published up to 15 February 2022 were identified. Medical Subject Heading (MeSH) terms such as “uterine cervical neoplasms” and “gene expression” were used as the keywords in all fields. Synonyms for the keywords were generated through MeSH terms from the Cochrane Library. Additional text terms were found by assessing the collected review articles. The search strategy involved a combination (“AND”) of the following sets of keywords: (1) “uterine cervical neoplasms” OR “cervical intraepithelial neoplasia” OR “cervical cancer” and (2) “gene expression.” Additional references were identified from the bibliographies of the retrieved studies.

### 2.2. Inclusion Criteria

Case–control, cross-sectional, and prospective observational studies with abstracts investigating the DEGs in the cervical tissue of cervical cancer patients were included. Only clinical studies using the microarray method to identify differentially expressed genes (DEGs) in the cervical tissue of cervical cancer patients were included to ensure the homogeneity of the data. Studies reporting a list of DEGs were included in this systematic review.

### 2.3. Exclusion Criteria

Publications without primary data, such as editorials, case reports, conference proceedings, and narrative review articles, were excluded. In silico, in vitro, and in vivo studies were excluded. Intervention studies for cervical cancer treatment and those using blood and non-cervical tissue samples were excluded. Studies that utilized other gene-expression methods, such as quantitative PCR, were not selected to avoid bias in gene selection. Next-Generation Sequencing (NGS) studies were excluded to avoid allele expression biases. Studies that did not include the list of DEGs in their report, Appendix A, or links to any other sources were also excluded.

### 2.4. Screening of Articles for Eligibility

Three phases of screening were performed on articles recovered from all resources. Duplicates were removed, and all articles with non-relevant titles were excluded in the first phase. The abstracts of the remaining articles were examined, and articles that did not meet the inclusion criteria were excluded from the second phase. Finally, the full texts of the remaining articles were reviewed thoroughly. Systematic reviews; meta-analyses; in vitro, in vivo, and in silico articles; and articles that did not meet the inclusion criteria were excluded in this third phase. All the authors were involved in the screening, selection, and data extraction phases. Figure 1 shows the PRISMA flow diagram summarizing the article-sorting process, and the reasons for article elimination.

### 2.5. Data Extraction

Data were extracted from the studies that fulfilled the inclusion criteria. All the authors participated in extracting the data. A data collection form was used to standardize the data collection, and all data extraction was performed independently. Any disagreements were discussed, decisions were made based on the majority consensus, and records of reasons for rejection were kept. The collected data are as follows: (1) author’s name, (2) article title, (3) study design, (4) sample size, (5) type of sample, (6) gene-expression method, (7) list of DEGs, and (8) conclusion.

### 2.6. Study Quality

The study quality of each paper was evaluated independently by NAH and MHE using the Joanna Briggs Institute critical appraisal tools [13]. The results of the quality of the study were validated by MHE and NAH. An overall score of less than 50% was used to rate a paper as a low-quality (high risk of bias) paper. If the overall score was 50–69%, it was rated as a moderate-quality (moderate risk of bias) paper, and if the overall score was more than 69%, it was rated as a high-quality (low risk of bias) paper.

### 2.7. Venn Diagram Analysis

All the DEGs were extracted and listed according to the studies. Then, a Venn diagram analysis was carried out using Bioinformatics & Evolutionary Genomics software [14]. The Venn diagram analysis was performed to identify the common DEGs between studies. Differentially expressed genes in at least two studies were selected for further analysis.

### 2.8. Protein–Protein Interaction (PPI) Network, Clustering, and Visualization

All selected DEGs from the Venn diagram analysis were pooled and analyzed through a PPI functional-enrichment analysis via STRING software (version 11.5, STRING Consortium 2022, Zurich, Switzerland, https://string-db.org/ accessed on 28 October 2022) to identify the PPI network [15]. The results from STRING were exported into Cytoscape software (version 3.8.0, Cytoscape Consortium, San Diego, CA, USA, http://www.cytoscape.org/, accessed on 4 November 2022) to visualize the molecular interaction networks and to integrate the gene-expression profiles of the DEGs [16]. A module analysis of the target network and protein clustering were performed using the Cytoscape MCODE plug-in (degree cut-off = 2, node score cut-off = 0.2, node density cut-off = 0.1, K-score = 2, and max depth = 100). The significantly enriched gene ontology was identified by analyzing the list of genes in each cluster using DAVID software (version 2021, DAVID Bioinformatic team, Frederick, MD, USA).

### 2.9. Gene Ontology (GO) and Pathway Enrichment Analysis

All the genes in each cluster were analyzed using the Database for Annotation, Visualization, and Integrated Discovery (DAVID) to discover the gene ontology that exhibited significant functional annotation enrichment related to cervical cancer pathogenesis [17]. Finally, the Kyoto Encyclopedia of Genes and Genomes (KEGG) pathway was utilized to expose the involvement of genes in the pathway related to cervical cancer [18].

## 3. Results

The four databases generated 4027 potentially relevant studies from the keywords. Based on the title, 307 duplicates were removed, and the abstracts of the other 3720 were screened. Upon screening the abstracts, 3681 papers were excluded. The full text of the other 24 papers was then retrieved. After a thorough review of the full text, 18 articles that did not meet the inclusion and exclusion criteria were excluded. Finally, six studies were selected for this systematic review. These six studies were published between the years 2006 and 2021. Prevention of sampling bias was assured through homogenized sampling by adhering strictly to the specific inclusion and exclusion criteria. Only studies that performed microarray analyses were chosen, and the sample size was between 2 and 25. Table 1 shows a summary of the characteristics of the included studies.

### 3.1. Patient Recruitment and Sample Collection Details

Sample collection was described in all studies. Kim et al. (2013), Miyatake et al. (2007), and Wong et al. (2006) stated that the staging of cervical cancer was carried out according to the International Federation of Gynaecology and Obstetrics (FIGO) criteria [20,22,24]. However, not all studies elaborated on the demographic profile of their patients. While the other five studies collected biopsies, Rajkumar et al. (2011) collected the archive-extracted total RNA from the biobank [21]. Normal cervical tissue biopsies were collected from women who underwent hysterectomy for other gynecological-related problems [19,21,24], from uterine leiomyoma patients [20], and from normal tissue adjacent to cervical squamous epithelia [22,23].

### 3.2. Study Quality

A detailed quality assessment of the included studies is listed in Appendix A. The included studies comprised four high-quality (low risk of bias) studies and two moderate-quality (moderate risk of bias) studies.

### 3.3. Identification of DEGs in Cervical Cancer

After removing duplicates, 1128 DEGs were extracted from all the selected studies. In the studies by Annapurna et al. (2021), Kim et al. (2013), Rajkumar et al. (2011), Miyatake et al. (2007), and Wong et al. (2006), only genes with an expression level of more than two or less than a negative two-fold change are identified as DEGs [19,20,21,22,24]. However, in the study by Gius et al. (2007), instead of using the fold change as a parameter to select DEGs, this study uses the calculated false discovery rate [23]. A list of 104, 409, 66, 22, 536, and 102 DEGs were reported in Annapurna et al. (2021), Kim et al. (2013), Rajkumar et al. (2011), Miyatake et al. (2007), Gius et al. (2007), and Wong et al. (2006), respectively [19,20,21,22,23,24].

### 3.4. Common DEGs among the Studies Identified via Venn Diagram Analysis

The common DEGs between studies were identified, in which *CDH3* and *CDKN2A* were differentially expressed in four studies (Annapurna et al., 2021; Gius et al., 2007; Rajkumar et al., 2011; Wong et al., 2006) [19,21,23,24]. Four other genes (*BST2, PLSCR1, SPINK5, PLOD2*) were common in three studies, and sixty-two genes were commonly expressed in two studies. Figure 2 shows the Venn diagram result for the DEGs from all the included studies. All 68 genes differentially expressed in at least two studies (Appendix A) were selected for further analysis.

### 3.5. Identification of Key Candidate Genes and Pathways via Protein–Protein Interaction (PPI) Network and Modular Analysis

Sixty-eight proteins from the selected DEGs were filtered into a PPI network complex containing 68 nodes and 181 edges with a PPI enrichment *p*-value of <1 × 10^−16^ and an average local clustering coefficient of 0.602. The network’s data were transferred from STRING to Cytoscape software to visualize the molecular interaction networks. By utilizing the Molecular Complex Detection Algorithm (MCODE), three significant modules from the PPI network complex were discovered. Figure 3 shows the three significant clusters created from the PPI network complex generated from the DEGs in cervical cancer patients. The functional annotation clustering shows that cluster 1 comprises 20 nodes and 58 edges (score = 6.105), while cluster 2 consists of 7 nodes and 13 edges (score = 4.33), and cluster 3 consists of 4 nodes and 6 edges (score = 4).

### 3.6. Gene Ontology (GO) and Pathway Enrichment Analysis of the Identified Clusters

The GO and pathway enrichment analyses showed that the DEGs in cluster 1 were primarily located in the nucleoplasm and cytosol. These DEGs in cluster 1 involve the cell cycle and regulation of transcription. Their molecular functions include protein binding, DNA binding, ATP binding, and transcription factor binding. The DEGs in cluster 2 are primarily located in the cytoplasm and regulate cell adhesion, migration, and inflammatory and immune responses. The molecular function of DEGs in cluster 2 includes extracellular matrix binding, cytokine activity, and fibronectin binding.

The DEGs in cluster 3 were mainly situated in the cytosol and cornified envelope. These DEGs are involved in protein heterotetramerization, peptide cross-linking, and epithelial cell differentiation. The molecular function of DEGs in cluster 3 includes structural constituents of the epidermis. Table 2 summarizes the functional annotation clustering of the DEGs.

### 3.7. Selection of Candidate Genes in Cervical Cancer Pathogenesis

From the PPI network and the functional annotation clustering, eight genes were chosen as the candidate genes based on their involvement in the relevant GO and their interaction with other genes in the PPI network through undirected first neighbor nodes (Figure 4). The candidate genes include *CDNK2A, VEGFA, PTGS2, MCM2, MCM4, MCM6, KRT1*, and *KRT10*. From the analysis, *CDNK2A, VEGFA*, and *PTGS2* interacted closely. Meanwhile, the *MCM* and *KRT* families interact less with other clusters. Further details were extracted for all the candidate genes. Their expression levels in cervical cancer samples and functional annotation of each gene set are shown in Table 3.

## 4. Discussion

Understanding the comprehensive molecular mechanism of cervical cancer pathogenesis is crucial for developing better screening, management, and treatment for cervical cancer patients. Thus, many gene-expression studies on cervical cancer via various methods have been reported [9,25,26]. This systematic review comprehensively explored the contribution of gene expression and interaction in cervical cancer. From the six selected studies that reported on the DEGs in cervical samples of cervical cancer patients, 68 genes were differentially expressed in at least two studies. The PPI network and modular analysis revealed three significant clusters. All clusters showed high node connection and interaction, in which cluster 1 comprised 20 nodes and 58 edges, cluster 2 consisted of 7 nodes and 13 edges, and cluster 3 consisted of 4 nodes and 6 edges.

From the functional annotation clustering analysis, the primary molecular mechanism in cervical cancer is divided into three main mechanisms according to the clusters. The first mechanism, which includes the DEGs in cluster 1, is primarily involved in the cell cycle pathway and specifically in regulating the transcription process via ATP binding, single-stranded DNA binding, and transcription factor binding. The second mechanism, which includes the DEGs in cluster 2, is mainly involved in cell signaling that regulates cell proliferation, adhesion, and migration in response to infection and inflammation. The third mechanism, which includes the DEGs in cluster 3, mainly involves cell differentiation via the regulation of the keratin filament network through protein heterotetramerization.

The individual genes in each cluster were further analyzed, and seven candidate genes were selected; these were *CDKN2A, VEGFA, PTGS2, MCM2, MCM4, MCM6, KRT1,* and *KRT10*. The CDKN2A protein is located in the nucleus and cytosol in cells (GO:0005634; GO:0005829) and is highly expressed in fat, testis, and adrenal tissue [27]. In cervical tissue, the expression of *CDKN2A* is typically low, but higher expression (6-fold to 342-fold change) was reported in cervical cancer tissue in four of the selected studies in this systematic review [19,21,23,24]. However, this finding is contradictory to an in vitro study by Luan et al. (2021) that reported on the low expression of *CDKN2A* in the cervical cancer cell lines and that the overexpression of *CDKN2A* inhibits cell proliferation and invasion of cervical cancer cells by arresting the cell cycle in the G1 phase [28]. Another contradictory finding was reported in an epigenetics study, in which the cervical cancer group showed significantly higher *CDKN2A* methylation than the control group [29], suggesting a low *CDKN2A* expression level in cervical cancer. Based on the significant interactions of *CDKN2A* with transcription factors, signaling molecules, and miRNAs, this gene has been proposed as a biomarker for cervical cancer prognosis [30]. Thus, due to the significantly promising role of *CDKN2A* in cervical cancer, further studies are crucial to address this conflicting finding.

The VEGFA protein is located in the cytoplasm of a cell and is highly expressed in thyroid, prostate, lung, and endometrium tissues [27]. In normal cervical tissue, the expression of *VEGFA* and its isoform (*VEGF165*) is low but is upregulated in cervical cancer tissue [24,31]. *VEGFA* expression impacts cervical cancer cells’ apoptosis, proliferation, migration, and invasion [32]. Thus, the inhibition of *VEGFA* expression could lead to the inhibition of cell migration and invasive motility in cervical cancer cells [33] while the *PTGS2* (*COX2*) protein, located in the cytoplasm, is downregulated in cervical cancer tissue [20,22]. However, the overexpression of *PTGS2* has been associated with a poor prognosis and resistance to cytotoxic therapy [34]. Several signaling mechanisms for regulating *PTGS2* in cervical cancer have been validated, including the EGF and nuclear factor κB (NF-κB) pathways [35], and PAR2 through an EGFR-dependent mechanism [34].

*MCM2, MCM4*, and *MCM6* are from the minichromosome maintenance complex (MCM) protein family, which are essential components of the pre-replicative complex for the initiation of DNA replication and cell division [36]. In cervical cancer, *MCM2* [24] and *MCM6* are upregulated while *MCM4* is downregulated [20]. However, the higher expression of *MCM2, MCM4*, and *MCM6* is significantly associated with favorable overall survival in cervical cancer patients [37].

*KRT1* and *KRT10* are from the cytokeratin gene family, which is highly expressed in the upper layer of the epidermis and the endothelial cells [38]. In cervical cancer, *KRT1* (type II; “acidic” keratin) and *KRT10* (type I; “basic” keratin) are upregulated in cervical cancer [21]. Thus, it is suggested that *KRT1* and *KRT10* are involved in the proliferation of cervical keratinocytes in cervical cancer.

Unlike the *MCM* family and *KRT* family, which interact closely within the cluster, *CDKN2A, VEGFA*, and *PTGS2* show broader interactions when incorporating proteins from other clusters. Interestingly, the signal transducer and activator of transcription 1 (*STAT1*) is found to interact with these three genes closely. *STAT1* is a tumor suppressor gene that plays an essential role in apoptotic and anti-apoptotic signaling [39]. From the functional annotation analysis, *CDKN2A*, *VEGFA, PTGS2*, and *STAT1* were involved in cancer pathways (hsa05200; KEGG pathway). The combination and interaction of these genes in cancer pathogenesis cover the major signaling pathways in cancer, such as proliferation, angiogenesis, cell cycle, and apoptosis pathways.

Major signaling pathways in proliferation, cell cycle, migration, apoptosis, and DNA repair are primarily involved in cancer development [40,41,42]. The phosphatidylinositol 3-kinase/protein kinase B/mammalian target of rapamycin (PI3K/AKT/mTOR) signaling pathway involved in cell proliferation, survival, invasion, migration, apoptosis, glucose metabolism, and DNA repair is associated with breast cancer pathogenesis [43]. Meanwhile, the JAK/STAT pathway, PI3K/Akt/mTOR pathway, Ras/Raf/MAPK pathway, and Wnt/β-catenin pathway contributed to hepatocellular carcinoma by regulating the cell growth, differentiation, apoptosis, and survival [44].

Besides the major signaling pathways usually exhibited by most cancers, cervical cancer also exhibits its unique pathways. From the functional annotation clustering performed, cervical cancer was also found to involve pathways related to infections such as human papillomavirus infection (hsa05165), human cytomegalovirus infection (hsa05163), human T-cell leukemia virus 1 infection (hsa05166), and staphylococcus aureus infection (hsa05150). Pathways that respond to infection such as NOD-like receptor-signaling pathway (hsa04621) and Toll-like receptor-signaling pathway (hsa04620) are also involved in cervical cancer.

The strength of this systematic review is that it includes studies from various cohorts and populations. Thus, a more comprehensive molecular mechanism for cervical cancer with reduced effects of genetic variations between populations can be identified. The homogeneity and bias reduction of the data were assured by only incorporating the DEGs from the microarray result. The integrated bioinformatic analysis of the DEGs pooled from the systematic review helps to better interpret the genes’ function and involvement in the molecular mechanism of cervical cancer.

Nevertheless, the main limitation of this systematic review is the incomplete listing of differentially expressed genes in some of the studies and the various statistical analyses used by the included studies in identifying the DEGs. However, the homogeneity of the data was established by adhering to the strict inclusion criteria and by selecting only DEGs detected by microarray. This systematic review added new insight into the molecular mechanism of cervical cancer. It will be beneficial to elucidate further downstream mechanisms of the DEGs by using RNA interference or gene knockdown strategies.

## 5. Conclusions

The molecular interactions in the cervical tissue of cervical cancer patients required extensive investigation to identify a comprehensive molecular mechanism in cervical cancer pathogenesis. The integrated bioinformatic analysis from this systematic review reveals three major molecular mechanisms involved in cervical cancer pathogenesis: cell cycle, cell differentiation, and infection. *CDKN2A, VEGFA, PTGS2, MCM2, MCM4, MCM6, KRT1, KRT10*, and *STAT1* were identified as key genes that regulate cervical cancer pathogenesis and progression. These genes can be utilized to develop a biomarker panel for screening cervical cancer and can become targeted genes for developing better treatment strategies for cervical cancer.

## Figures and Tables

**Figure 1 cancers-15-00853-f001:**
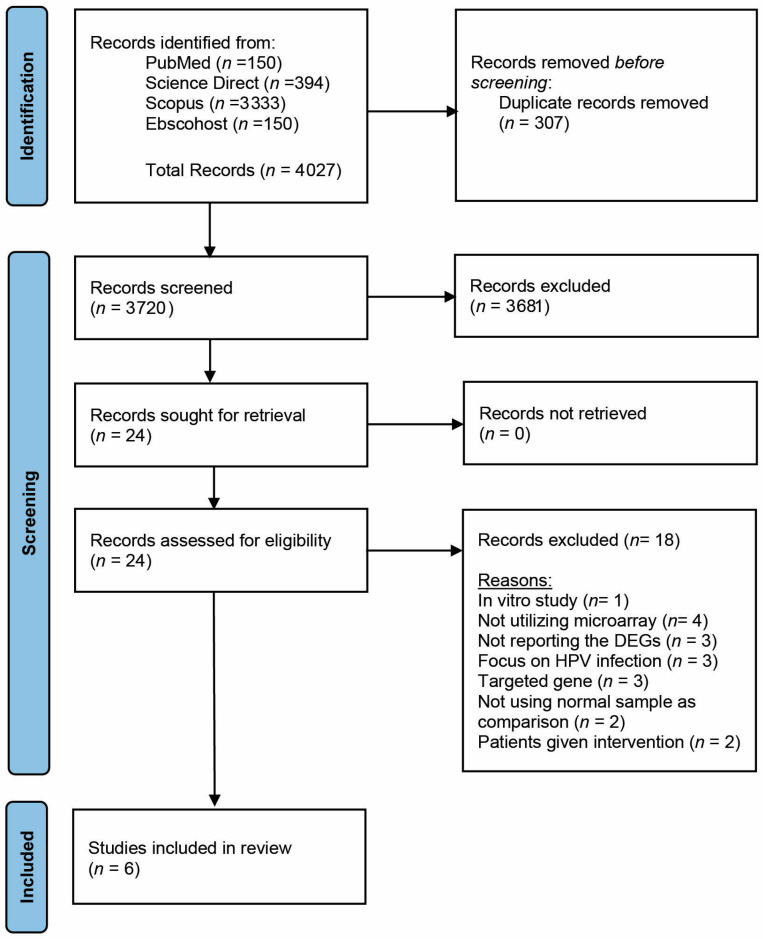
PRISMA flow diagram for studies’ selection in this systematic review.

**Figure 2 cancers-15-00853-f002:**
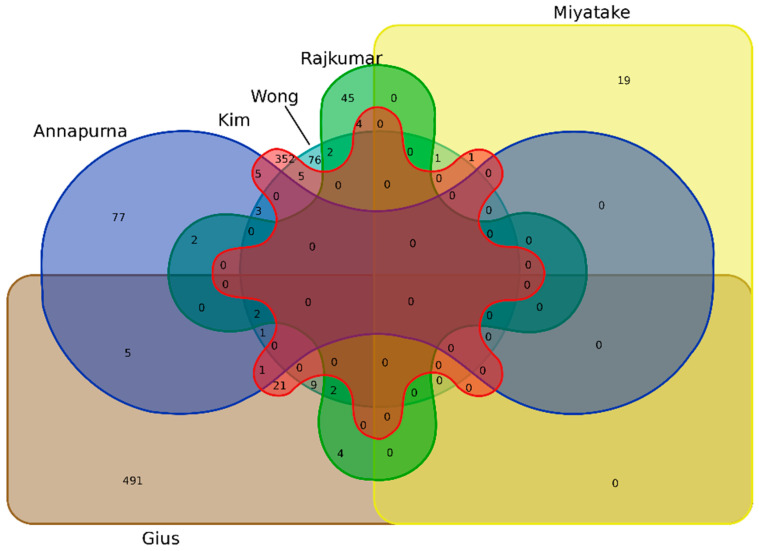
Venn diagram of the DEGs from all the selected studies.

**Figure 3 cancers-15-00853-f003:**
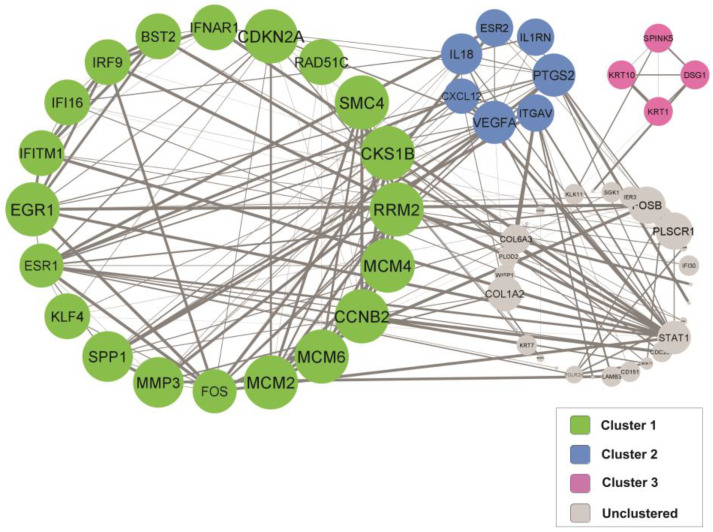
PPI network of the DEGs collected from the selected studies.

**Figure 4 cancers-15-00853-f004:**
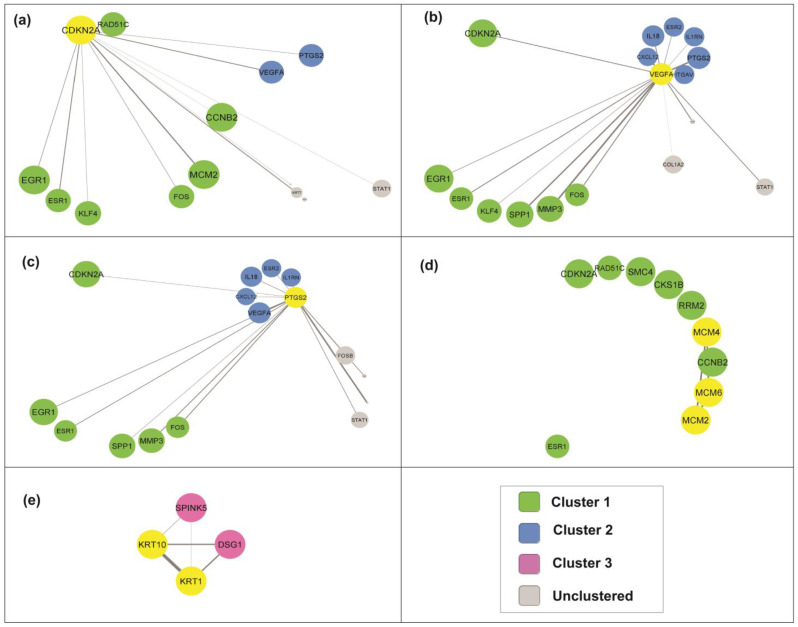
Protein interaction through undirected first neighbor of the candidate genes (highlighted in yellow) from cluster 1: (**a**) *CDKN2A*, (**b**) *VEGFA*, (**c**) *PTGS2*, (**d**) *MCM* family, and (**e**) *KRT* family.

**Table 1 cancers-15-00853-t001:** Summary of the characteristics of the included studies.

Author, Year (References)	Country	Study Design	Sampling	Sample Size (*n*)	Gene-Expression Analysis	Upregulated Genes	Downregulated Genes
Annapurna et al., 2021 [19]	India	Case–control	Cervical tissue	7 patients, 3 normal	Microarray	78	26
Kim et al., 2013 [20]	Korea	Case–control	Cervical tissue	28 patients, 17 control	Microarray	208	201
Rajkumar et al., 2011 [21]	India	Case–control	Cervical tissue	28 patients, 5 normal	Microarray	47	19
Miyatake et al., 2007 [22]	Japan	Cross-sectional	Cervical tissue	2 patients	Microarray	8	14
Gius et al., 2007 [23]	USA	Cross-sectional	Cervical tissue	85 patients	Microarray	536	
Wong et al., 2006 [24]	Hong Kong	Case–control	Cervical tissue	29 patients, 18 controls	Microarray	19	83

**Table 2 cancers-15-00853-t002:** Functional annotation clustering of the DEGs with highlighted candidate genes.

Cluster	Term	Description	Genes	*p*-Value
1	CC_ GO:0005654	Nucleoplasm	*EGR1, CDKN2A, FOS, KLF4, SMC4, ESR1, CKS1B, RAD51C, IFI16, MCM4, MCM6, IRF9, MCM2*	6.06 ×10^−6^
	CC_GO:0005829	Cytosol	*BST2, CCNB2, RAD51C, IFI16, CDKN2A, FOS, KLF4, SMC4, ESR1, IRF9*	1.96 × 10^−2^
	CC_GO:0000785	Chromatin	*EGR1, FOS, KLF4, ESR1, IRF9, MCM2*	1.32 × 10^−3^
	BP_GO:0045944	Positive regulation of transcription from RNA polymerase II promoter	*EGR1, IFI16, CDKN2A, FOS, KLF4, ESR1, IRF9*	3.96 × 10^−4^
	BP_GO:0007049	Cell cycle	*CDKN2A, MCM4, MCM6, SMC4, MCM2, CKS1B*	1.11 × 10^−5^
	BP_GO:0045893	Positive regulation of transcription, DNA-templated	*EGR1, CDKN2A, SPP1, FOS, KLF4, ESR1*	3.13 × 10^−4^
	MF_ GO:0005524	ATP binding	*RAD51C, MCM4, MCM6, SMC4, MCM2*	4.50 × 10^−2^
	MF_ GO:0003697	Single-stranded DNA binding	*MCM4, MCM6, SMC4, MCM2*	1.41 × 10^−4^
	MF_GO:0008134	Transcription factor binding	*IFI16, CDKN2A, FOS, ESR1*	8.12 × 10^−4^
	hsa04110	Cell cycle	*CCNB2, CDKN2A, MCM4, MCM6, MCM2*	8.53 × 10^−5^
2	CC_GO:0005737	Cytoplasm	*IL1RN, CXCL12, IL18, PTGS2, VEGFA*	0.049
	BP_ GO:0045785	Positive regulation of cell adhesion	*CXCL12, ITGAV, VEGFA*	1.37 × 10^−4^
	BP_GO:0030335	Positive regulation of cell migration	*CXCL12, ITGAV, VEGFA*	0.002
	BP_ GO:0006954	Inflammatory response	*IL1RN, IL18, PTGS2*	0.006
	MF_ GO:0005125	Cytokine activity	*IL1RN, IL18, VEGFA*	0.002
	MF_ GO:0050840	Extracellular matrix binding	*ITGAV, VEGFA*	0.009
	hsa05165	Human papillomavirus infection	*ITGAV, PTGS2, VEGFA*	0.022
3	CC_GO:0005829	Cytosol	*KRT1, SPINK5, DSG1, KRT10*	0.019
	CC_ GO:0045095	Keratin filament	*KRT1, KRT10*	0.015
	BP_ GO:0051290	Protein heterotetramerization	*KRT1, KRT10*	0.002
	BP_ GO:0018149	Peptide cross-linking	*KRT1, KRT10*	0.005
	MF_ GO:0030280	Structural constituent of epidermis	*KRT1, KRT10*	0.006

**Table 3 cancers-15-00853-t003:** The expression regulation of candidate genes and their functional annotations.

Candidate Gene		Functional Annotation	
Upregulated	Downregulated	Term	Description
*CDKN2A*		GO:0000079	Regulation of cyclin-dependent protein serine/threonine kinase activity
		hsa01522	Endocrine resistance
		hsa05166	Human T-cell leukemia virus 1 infection
		GO:0008134	Transcription factor binding
		hsa05200	Pathways in cancer
		hsa04110	Cell cycle
*VEGFA*	*PTGS2*	hsa04370	VEGF signaling pathway
		GO:0071456	Cellular response to hypoxia
		hsa05165	Human papillomavirus infection
		GO:0043154	Negative regulation of cysteine-type endopeptidase activity involved in apoptotic process
*MCM2, MCM6*	*MCM4*	GO:0006270	DNA replication initiation
		hsa03030	DNA replication
		GO:0003678	DNA helicase activity
		GO:0000727	Double-strand break repair via break-induced replication
*KRT1*	*KRT10*	GO:0030280	Structural constituent of epidermis
		GO:0018149	Peptide cross-linking
		113800	Epidermolytic hyperkeratosis

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
