# Peer review of "Candidate Genes and Pathways in Cervical Cancer: A Systematic Review and Integrated Bioinformatic Analysis"

_cancers, 2023, doi:10.3390/cancers15030853_

Round 1
Reviewer 1 Report
1) Human gene symbols should be italicized. Please correct them throughout the manuscript and supplementary materials.
2) In Figure 1, please align the direction and length of arrows as well as boxes.
3) Please check and use the decimal comma with metric units in the manuscript. E.g. 4027 in line 21
4) Figure 2, please increase the size of numbers in the Venn diagram of DEGs
5) Should the microarray be included in the title or mentioned in the abstract since this systematic review only incorporated the DEGs from the microarray results?
Author Response
REBUTTAL
We appreciate the suggestions and comments given by the honourable reviewer. We are happy to comply with all suggestions and comments. All changes in the revised version of the manuscript appear in RED colour.
Response to Reviewer 1 Comments
Point 1: Human gene symbols should be italicized. Please correct them throughout the manuscript and supplementary materials.
Response 1: We apologise for the error. All human gene symbols have been italicised in red throughout the manuscript and supplementary materials.
Point 2: In Figure 1, please align the direction and length of arrows as well as boxes.
Response 2: The direction and length of arrows in Figure 1 as well as boxes have been aligned accordingly.
Point 3: Please check and use the decimal comma with metric units in the manuscript. E.g. 4027 in line 21.
Response 3: The decimal comma with metric units have been corrected:
Line 21: 4,027
Line 24: 1,128
Figure 1: Numbers in the PRISMA flow diagram
Line 145: 4,027
Line 146: 3,720
Line 170: 1,128
Point 4: Figure 2, please increase the size of numbers in the Venn diagram of DEGs
Response 4: We apologise for the size of number. However, the figure was generated from the software and the size of numbers could not be changed.
Point 5: Should the microarray be included in the title or mentioned in the abstract since this systematic review only incorporated the DEGs from the microarray results?
Response 5: Thank you for the suggestion. We think that the word 'microarray' is good enough to be mentioned in the abstract and does not look good scientifically if included in the title. Hope the reviewer accepts our humble request.

Reviewer 2 Report
The authors have described a very systematic approach and review to identify new gene candidates for screening and treatment of cervical cancer. I have two specific comments regarding their study;
1. The authors report that they selected only microarray based articles for their study so that biases of methods would not be introduced. However, given the various uncertainties associated with the entire sample to answer process of these types of studies, I am not convinced that eliminating other methods like transcriptomics or targeted gene amplifications is sound. In order to obtain unequivocal evidence for specific genes, it would be useful to include other studies particularly if the same genes are being seen in diverse studies. There is more value to identifying such candidates.
2. In the discussion section, I would like to see more comparison to what has already been discovered in past studies with respect to both the genes and the specific pathways. Similarly, there should be some commentary on pathways that have been identified in other cancers - is there anything unique to cervical cancer? Can there be genes/pathways targeted for any cancer?
Author Response
REBUTTAL
We appreciate the suggestions and comments given by the honourable reviewer. We are happy to comply with all suggestions and comments. All changes in the revised version of the manuscript appear in RED colour.
Response to Reviewer 2 Comments
Point 1: The authors report that they selected only microarray based articles for their study so that biases of methods would not be introduced. However, given the various uncertainties associated with the entire sample to answer process of these types of studies, I am not convinced that eliminating other methods like transcriptomics or targeted gene amplifications is sound. In order to obtain unequivocal evidence for specific genes, it would be useful to include other studies particularly if the same genes are being seen in diverse studies. There is more value to identifying such candidates.
Response 1: Thank you for the valuable insight. Papers that utilised the transcriptomics or targeted gene amplification method were excluded to ensure a homogenised data set. Targeted gene amplifications method such as qPCR have different levels of sensitivity in measuring the gene expression compared to microarray. The expression level of the targeted genes also could not be compared with the other genes in the genome as it only measures specific genes, making the result incomparable with microarray. Other transcriptomic studies such as NGS were not excluded due to the different levels of sensitivity in measuring the gene expression compared to microarray and to avoid allele expression biases.
Point 2: In the discussion section, I would like to see more comparison to what has already been discovered in past studies with respect to both the genes and the specific pathways. Similarly, there should be some commentary on pathways that have been identified in other cancers - is there anything unique to cervical cancer? Can there be genes/pathways targeted for any cancer?
Response 2: Thank you for the precious suggestion. We have looked into our data again and improved the discussion according to the comments given. (Line 301-308)
Major signaling pathways in proliferation, cell cycle, migration, apoptosis and DNA repair are primarily involved in cancer development [41-43]. The phosphatidylinositol 3-kinase/protein kinase B/mammalian target of rapamycin (PI3K/AKT/mTOR) signaling pathway involved in cell proliferation, survival, invasion, migration, apoptosis, glucose metabolism and DNA repair is associated with breast cancer pathogenesis [44]. While JAK/STAT pathway, PI3K/Akt/mTOR pathway, Ras/Raf/MAPK pathway and Wnt/β-catenin pathway contributed to hepatocellular carcinoma by regulating the cell growth, differentiation, apoptosis and survival [45].
Round 2
Reviewer 2 Report
Please edit the statement in your exclusion criteria section 2.3 to reflect your response as it says "NGS" not targeted amplification.
"Studies that utilized other gene expression methods, such as Next-Generation Sequencing (NGS) and quantitative PCR were not selected to avoid bias in gene selection".
I would also provide the same rationale you have in your response to point 1 in this section.
Author Response
Point 1: Please edit the statement in your exclusion criteria section 2.3 to reflect your response as it says "NGS" not targeted amplification.
Response 1: We apologise for the unclear statement. The statement has been rephrased:
(Line 90-93).
"Studies that utilized other gene expression methods, such as quantitative PCR, were not selected to avoid bias in gene selection. Next-Generation Sequencing (NGS) studies were excluded to avoid allele expression biases"
Round 3
Reviewer 2 Report
No more comments at this time.